# Controlling single rare earth ion emission in an electro-optical nanocavity

Likai Yang [1], Sihao Wang[1], Mohan Shen[1], Jiacheng Xie [1] & Hong X. Tang [1] ✉

Rare earth emitters enable critical quantum resources including spin qubits, single photon sources, and quantum memories. Yet, probing of single ions remains challenging due to low emission rate of their intra-$4f$ optical transitions. One feasible approach is through Purcell-enhanced emission in optical cavities. The ability to modulate cavity-ion coupling in real-time will further elevate the capacity of such systems. Here, we demonstrate direct control of single ion emission by embedding erbium dopants in an electro-optically active photonic crystal cavity patterned from thin-film lithium niobate. Purcell factor over 170 enables single ion detection, which is verified by second-order autocorrelation measurement. Dynamic control of emission rate is realized by leveraging electro-optic tuning of resonance frequency. Using this feature, storage, and retrieval of single ion excitation is further demonstrated, without perturbing the emission characteristics. These results promise new opportunities for controllable single-photon sources and efficient spin-photon interfaces.

Quantum networks benefit from solid-state spins for their potential roles as quantum memories[1,2] and coherent transducers[3,4]. Addressing single atomic defect is also a key task in implementing solid-state qubit[5], spin-photon entanglement[6,7], and single photon source[8]. Among various systems studied, rare earth ion (REI) stands out thanks to its narrow, highly coherent transitions in both optical and microwave domains[9,10]. In particular, erbium (Er) ion draws great attention due to its telecom-band emission. However, addressing weakly coupled $4f$–$4f$ optical transition of single REIs remains challenging. A critical solution to this is through Purcell enhancement of emission rate in optical cavities[11]. Using this method, optical probing of single REIs is recently achieved by coupling micro-cavities with bulk host crystals: individual Er ion in yttrium orthosilicate (YSO) is addressed in silicon nanocavities stamped on YSO[12] and cryogenic Fabry-Perot resonators[13]; single neodymium (Nd) ion is also resolved in nanophotonic cavities directly fabricated on yttrium orthovanadate (YVO) crystal using focused ion beam[14].

Toward better control and scalability of REI coupled systems, integrated photonic resonators with separate and instant tunability are highly desirable. Dynamic control of ion-cavity coupling has been realized with Er-doped nanoparticles embedded in a piezoelectrically tunable Fabry-Perot microcavity[15] as well as ytterbium (Yb) doped lithium niobate microdisk resonators[16]. Real-time modulation of emission rate could be used to design tailored single photon source. Such capacity is also advantageous in building high-efficiency spin-photon interface. A target transition can be switched on-resonance when strong drive is needed and switched off by detuning the cavity[17]. It is also possible to modify spin cyclicity[18] by matching cavity frequency to different Zeeman transitions. As an example, branching ratio between optical transitions in an electronic spin Λ system can be controlled by selectively enhancing the spin-conserving or spin-flipping transitions with the cavity. This could enable efficient spin pumping and initialization, which is vital in both quantum memory and qubit control applications[19,20]. Furthermore, individual tunablity will be particularly helpful when integrating systems with multi-qubit operation[21].

Lithium niobate (LN) is a commonly used host crystal for REIs in both quantum and classical applications[22,23]. Meanwhile, it is also an important material in photonic technology with outstanding electro-optic[24], piezoelectric[25], and nonlinear[26] properties. With recent development on thin film lithium niobate on insulator (LNOI), on-chip optical resonators such as rings[27], microdisks[28], and photonic crystals[29] are proposed and fabricated. In particular, ring[30] and photonic crystal[31] cavities have been integrated with metal electrodes to develop electro-

[1]Department of Electrical Engineering, Yale University, New Haven, CT 06511, USA. ✉e-mail: hong.tang@yale.edu

optic modulators with high efficiency and bandwidth. Featuring high-quality factor, small mode volume, and electro-optical tunability, these resonators are favorable platforms to access REIs. To date, incorporation of REIs into LNOI-integrated photonics has been realized by ion implantation[32,33], flip chip bonding[34], and smart-cut technique[35]. The last, in particular, guarantees full overlap between REIs and optical mode while preserving bulk coherence properties[36]. Still, most of existing works focus on ion ensembles and single Er emitters in LN host are yet to be studied.

In this work, we demonstrate detection and control of single ion emission in smart-cut erbium-doped lithium niobate (ErLN) thin film. Photonic crystal nanobeam cavities with electro-optical tunability are fabricated to achieve a high Purcell enhancement of 177. By tuning the cavity to the tail of inhomogeneous broadening, single Er emission is spectrally resolved. It is then verified by second-order autocorrelation measurement yielding $g^2(0) = 0.38$, indicating majority of collected photons come from a single ion. We also show that the waveform of Er emission can be tailored using pulsed tuning voltage. The storage and retrieval of single ion excitation is further realized by detuning and re-aligning the cavity after the excitation pulse. The emission spectrum is shown not to be perturbed by this process. Our results pave the way for building tractable and scalable spin-photon interface with REIs.

## Results

### Device design and preparation

The devices are fabricated using 300 nm z-cut ErLN (100 ppm doped) thin films (from NanoLN), with a similar design concept previously used in developing LN electro-optic modulators[31]. Schematic drawing of the photonic crystal structure is shown in Fig. 1a. On a half-etched ridge waveguide, periodic holes are patterned to form photonic crystal mirror. Silicon dioxide (SiO₂) layer beneath the waveguide is removed with buffered oxide etch (BOE) that goes through the holes. Such suspended structure improves mode confinement, thus is necessary for reaching high-quality factor. Gold electrodes are then deposited along the waveguide for frequency tuning. Detailed fabrication process and device dimension is discussed in Supplementary Note 1. Er ions are uniformly distributed in LN and can be addressed via fluorescence emission. To form a cavity with a defect mode, lattice constant of the photonic crystal is tapered down in the middle, as plotted in Fig. 1b. One side of the mirror is also tapered so that the cavity can be accessed by measuring the reflectivity. The simulated electric field profile of the optical fundamental TE mode is illustrated in Fig. 1c, with field component mostly in the crystal y-direction. With an in-plane voltage bias, the electro-optic coefficient $r_{22} = 7$ pm/V[37] of LN is utilized. The half-etched slab helps to achieve larger tunability by increasing the overlap between optical mode and electric field. It also provides strong mechanical support to increase the fabrication yield.

Figure 1d shows false color SEM images of the actual device, with a zoom-in view of the waveguide and the apodized grating coupler[38] used for fiber-to-chip coupling.

To match Er emission wavelength, cavity resonance is swept in the fabrication process by varying the defect lattice constant. According to simulation, a 1 nm difference in lattice constant will result in -3 nm resonance shift. For our devices, we sweep across a 7 nm range, sufficient to overcome uncertainty caused by fabrication imperfection and film thickness variation. Cryogenic electrical and optical access to the devices are realized by wire bonding and fiber glue techniques, respectively. The packaged devices are then loaded at 1 K plate of a dilution refrigerator for cryogenic measurement.

### Device characterization

At cryogenic temperature, the reflection spectrum of our device is first measured, as shown in Fig. 2a. The resonance exhibits a quality factor of 158 k and extinction ratio close to 10 dB. The total throughput of our device is around 1%. Electro-optic frequency tuning is calibrated in Fig. 2b, by applying voltage up to 200 V. Tuning rate of 1.6 pm/V is extracted. For our device geometry, breakdown of lithium niobate will usually be observed with a voltage larger than 500 V. This gives a total tuning range of -1.5 nm by applying bipolar voltage within this range.

Fluorescence from Er ions is collected using a commercial superconducting nanowire single photon detector (SNSPD). In the experiment, we use a pulse measurement setup in which the optical pump, SNSPD bias, and tuning voltage can all be controlled with synchronized pulse sequences (see Supplementary Note 2 for details). With a resonant fluorescence scheme, population lifetime of Er ions in the waveguide and the cavity are extracted and fitted by a single exponential function, respectively. The results are plotted in Fig. 2c. The cavity enhancement of Er emission is shown by reduction of lifetime from $T_{wg} \approx 2.5$ ms to $T_{cav} \approx 14$ μs. This yields a Purcell factor $P = T_{wg}/T_{cav} - 1 = 177$. Theoretically, the Purcell enhancement of emission rate obeys the relation $P \propto Q/V_{mode}$[39]. Our device achieves strong enhancement by miniaturizing the mode volume $V_{mode}$ to -0.55 μm³ while maintaining high-quality factor ($Q$). The measured average Purcell factor is also in good agreement with simulated value of 150 (Supplementary Note 3).

### Single ion detection

Emission of Er $Y_1 - Z_1$ transition in LN is inhomogeneously broadened, centering at -1532 nm. Linewidth of the broadening can be measured by the fluorescence intensity in the waveguide at different wavelengths, as shown in Fig. 3a. A full width at half maximum (FWHM) of 160 GHz is extracted with a Gaussian fit, close to literature value of 180 GHz. Electro-optic frequency tuning provides a convenient and deterministic way to park the cavity resonance at different positions of

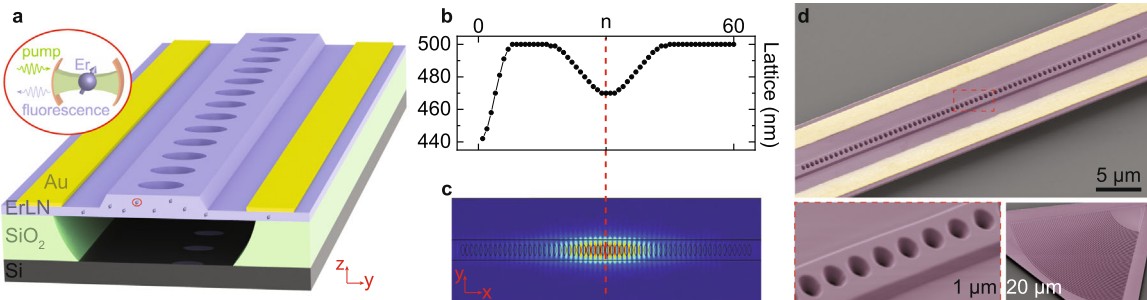

**Fig. 1 | Device design. a** Schematic drawing of our device (not to scale). Periodic holes are patterned on a suspended ridge waveguide to form photonic crystal mirror. Gold electrodes are deposited along the waveguide for electro-optic frequency tuning. Erbium (Er) ions are uniformly distributed in lithium niobate (LN). Ions in the photonic cavity can be addressed by collecting fluorescence. **b** Cavity design principle. The lattice constant is tapered down in the middle to support a defect mode and on one side for coupling. **c** Finite element simulation of resonance mode profile (fundamental TE). **d** False color SEM images of the actual device, with a zoom-in view on the waveguide. The grating coupler for fiber-to-chip coupling is also shown.

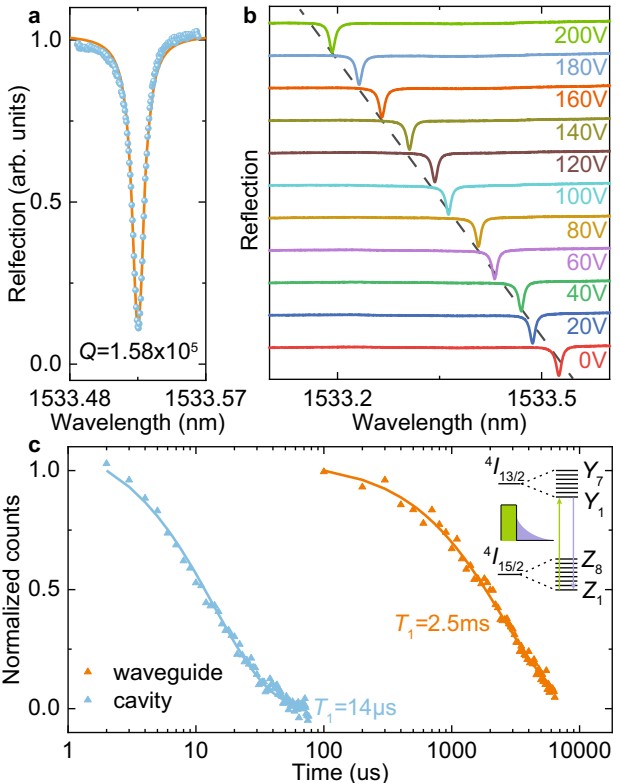

**Fig. 2 | Device characterization. a** Reflection spectrum of the resonator. A Lorentzian fit (orange curve) yields $Q = 1.58 \times 10^5$. **b** Resonance frequency shift under different applied voltage. The reflection spectrum of each voltage is offset for clarity. A tuning rate of 1.6 pm/V is extracted. **c** Time-domain fluorescence measurement of Er ions in the waveguide (orange) and cavity (blue) and exponential fitting. Enhancement of emission rate is shown by the shortened lifetime ($T_1$) inside the cavity. The inset shows an energy level diagram of Er ions. We pump (green) on the $Y_1 - Z_1$ transition centering around 1532 nm and collect the resonant fluorescence (purple).

inhomogeneous distribution. For the following measurement, a device with $Q \approx 100$ k is used. We first tune the cavity to 1533.274 nm and measure the spectrum of cavity enhanced fluorescence by sweeping the excitation laser across the resonance, as shown in Fig. 3b. The corresponding fluorescence intensity shows a continuous peak that decays with the detuning, suggesting that an ensemble of ions are excited. In order to resolve single ion emission, we need to tune further away to the tail of inhomogeneous broadening, where the ions are more dilute spectrally. The results are shown in Fig. 3c. The discrete peaks seen in the spectrum indicate that the fluorescence comes from single or several Er ions.

To confirm that the collected photons are from single ions, a second-order autocorrelation measurement is performed on one of the emission peaks (red star mark in Fig. 3c). Since we are measuring resonant fluorescence signal with one detector, a time-bin analysis is imposed. In the experiment, fluorescence signal in a 10 μs window after a 1 μs excitation pulse is collected, with a repetition rate of 20 μs. During data analysis, the photons collected after a single excitation pulse are pinned into one time bin. The coincidence events between time bins with offset $\tau$ are then extracted and normalized to calculate $g^{(2)}(\tau)$, while $g^{(2)}(0)$ corresponds to detecting two photons in a single time bin. The measured results are shown in Fig. 3d, with $g^{(2)}(0) = 0.38 \pm 0.08$. A simple signal-to-noise calculation of $g^{(2)}(0)$ (Supplementary Note 4) suggests that ~80% of the collected photons come from a single emitter. The remaining is attributed to SNSPD dark count (~10%) and background emission (~10%) from ions that are weakly coupled with the resonator, which are also the main factors that limit

$g^{(2)}(0)$. This is in good agreement with a measured count rate of ~200 Hz when at the emission peak and ~40 Hz at the background. Another instability comes from the spectral diffusion of the ion and frequency drift of the laser which is not actively stabilized. We account for a total drift in the order of MHz/min in our system, which is compensated by manually tuning the laser during the $g^{(2)}$ measurement. Such drift also contributes to the linewidth of single ion emission peaks in Fig. 3c, which ranges from 20 to 40 MHz.

To further improve quantumness of the emission (i.e., smaller $g^{(2)}(0)$), a resonator with frequency further away from the 1532 nm emission center can be used. This helps to suppress background emission by reducing the number of ions in the cavity. Also, the total transmission of our device is relatively low. With our design, transmission >10% can be routinely achieved at room temperature, but is reduced during cooldown due to fiber misalignment caused by thermal contraction. Optimization of fiber gluing or improved grating coupler design can boost the collection efficiency of single ion emission. Frequency locking of the laser would also be helpful to extend the available probing time.

**Electro-optic control of Er emission**

Tuning the cavity in and out of resonance with the ions modifies the coupling between them, thus changes the emission rate. We demonstrate this by pulsing the tuning voltage while collecting the fluorescence. The corresponding pulse sequence used in the experiment is shown in Fig. 4a. After exciting the ions on-resonance with the cavity using the pump pulse, the gate pulse for controlling the SNSPD switches on to collect fluorescence signal. Meanwhile, a 40 V bias voltage detunes the cavity so that Er emission into the cavity is suppressed, preserving the excited state population. The cavity can then be tuned back on resonance to restore ion-cavity coupling and enable the emission. This effect is clearly shown in Fig. 4b, demonstrating real-time control of emission intensity. The switching time of this process is mostly limited by the amplifier used for generating high voltage, which has a bandwidth of around 1 MHz.

With a similar protocol, storage and retrieval of single ion excitation can be achieved. To do so, we first focus onto a single ion emission peak. The lifetime of the ion is measured to be 10 μs, as shown by the red trace in Fig. 4d. A control sequence illustrated in Fig. 4c is then applied. Here, the cavity is detuned right after pump pulses to store ion excitation. The fluorescence is then collected in a 20 μs gate after tuning the cavity back after a certain delay. The storage lifetime of single ion excitation is extracted by varying the delay and the applied voltage, as shown in Fig. 4d. When 10 V bias is used, the cavity is not completely tuned away from the ion, so the lifetime extends to 172 μs. Larger tuning voltage of 80 V further recovers the ion lifetime to 2.82 ms, similar to waveguide value. We also note that the lifetime saturates around this value for larger tuning voltage. The slightly lengthened lifetime in detuned cavity could potentially be explained by the modification of spontaneous emission rate in photonic crystal band gap[40,41]. The fluorescence spectrum after 100 μs delay of cavity detuning is then measured and compared to the spectrum measured right after the pump pulse. The results are shown in Fig. 4e. Similar frequency and linewidth of the peaks in these two curves suggest that the storage and retrieval process does not perturb the emission spectrum or introduce significant spectral diffusion. The relative frequency offset could come from the system drift discussed above. The emission peaks for individual ions could experience different amount of random shift as the two spectrums are collected in sequence with time interval around 30 minutes. The red star mark indicates the point chosen for the storage measurement.

## Discussion

In conclusion, we have demonstrated the detection and electro-optic control of single Er emission in a LNOI photonic crystal nanobeam

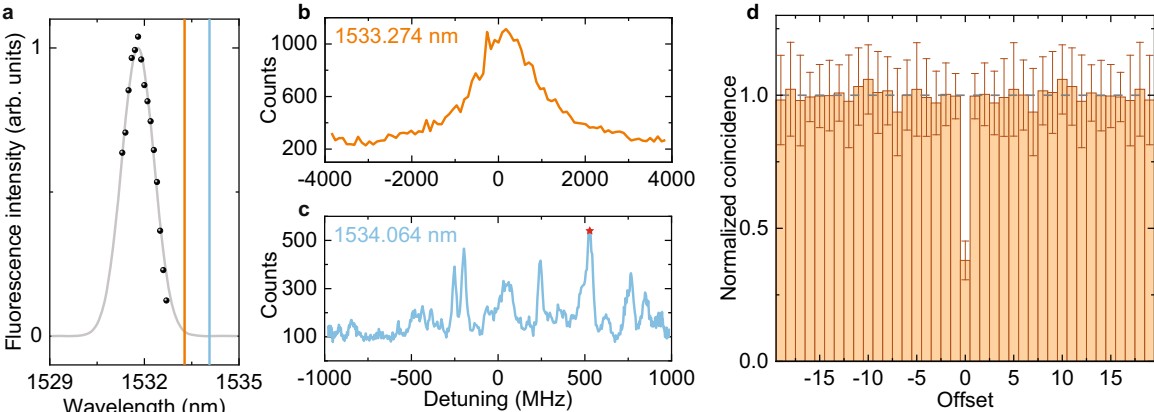

**Fig. 3 | Single ion detection. a** Inhomogeneous distribution of Er ions, measured by the fluorescence intensity in the waveguide. Gray curve is a Gaussian fit with FWHM of 160 GHz. Orange and blue lines indicate the frequency that the cavity is tuned to for fluorescence spectrum measurement shown in panels **b** and **c**. **b** Fluorescence spectrum inside the resonance when it is tuned to 1533.274 nm. A continuous emission spectrum shows that an ensemble of ions are probed. **c** Fluorescence spectrum when the resonance is tuned further to the tail of inhomogeneous broadening, at 1534.064 nm. Discrete peaks suggest single ion emission. The star mark indicates the point for $g^{(2)}$ measurement. **d** Second-order autocorrelation function $g^{(2)}$ for the single ion emission. The $x$ axis is the number of offset between excitation pulses, $y$ axis is normalized coincidence. Calculated value of $g^{(2)}(0) = 0.38 \pm 0.08$ proves that most of the collected photons come from a single ion. The data is symmetric around zero since a single detector is used. The error bars show the standard deviation of the measurement results.

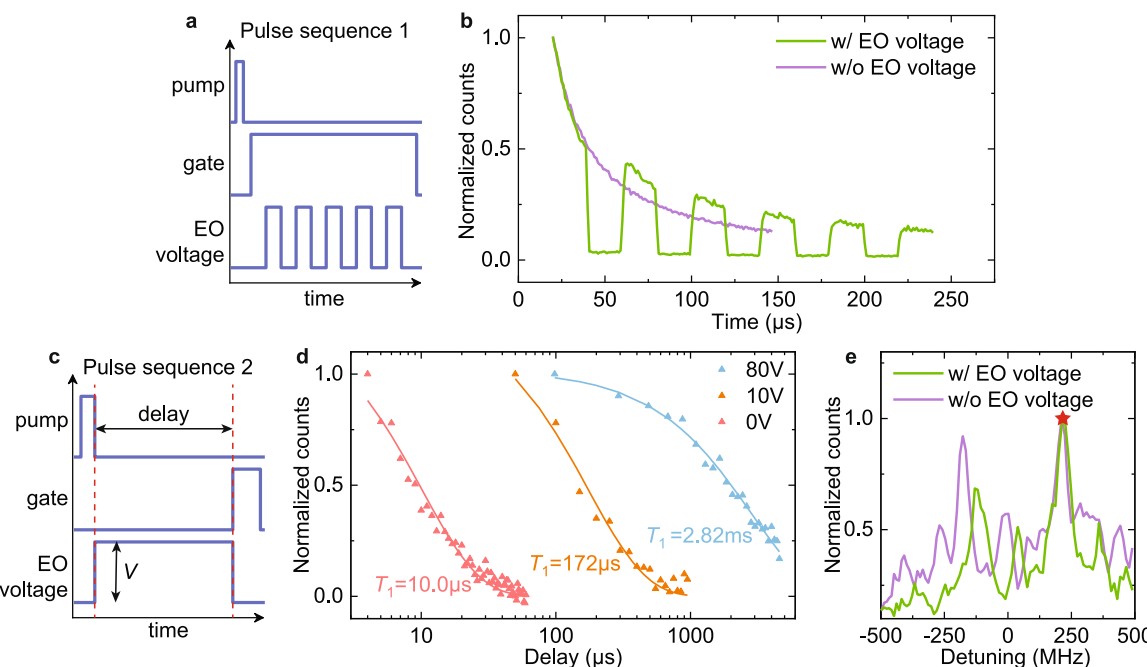

**Fig. 4 | Electro-optic control of Er emission. a** Pulse sequence used to demonstrate control of Er emission. Pump: excitation laser pulse; gate: on/off control of SNSPD for fluorescence collection; EO voltage: voltage sent to the device for frequency tuning. In this sequence, fluorescence signal is collected while modulating the cavity frequency to suppress and enhance Er emission. **b** Measured results with pulse sequence in a. When electro-optic (EO) voltage is applied (green curve), a pulsed Er emission can be seen. The decay when cavity is on resonance corresponds well with the case without EO voltage (purple curve). **c** Pulse sequence for storage and retrieval of single ion excitation. After exciting a single ion, the cavity is detuned for a certain delay to store the excitation. The emission is then retrieved by tuning the cavity back on resonance. **d** Measured results with pulse sequence in c. The lifetime ($T_1$) of single Er ion is first probed to be 10 µs, shown by the red curve. The storage scheme is then applied with different tuning voltage. With 10 V (orange) and 80 V (blue), an extended lifetime of 172 µs and 2.82 ms is realized, respectively. **e** Fluorescence spectrum measured right after the pump pulse (purple curve) and after 100 µs delay in the storage scheme (green curve). The single ion emission peaks match well for two cases. The relative shift is potentially caused by system drift including laser instability and spectral diffusion. The red star mark indicates the peak used for measurement in c.

cavity. Exhibiting high-quality factor and small mode volume, the resonator enhances the Er emission rate from a waveguide lifetime of 2.5 ms to 14 µs. Together with electro-optic tuning of resonance frequency to the tail of inhomogeneous broadening, single ion detection is enabled. A self-intensity autocorrelation of $g^{(2)}(0) = 0.38$ is measured, and practical ways to further improve single ion emission rate are discussed. The cavity is then shown to be able to store and shape single Er emission, with storage lifetime in detuned cavity matching the waveguide value. Utilizing electric field along $y$-direction, tuning rate of 1.6 pm/V is achieved. Larger tunability could be realized with a different geometry or film orientation. For example, the electro-optic coefficient $r_{33}$ along $z$-direction is five times larger than the $r_{22}$ we use

here. The advantage of using $r_{22}$ is that the DC stark shift of ErLN vanishes in this orientation[42], so that it could be separated from electro-optic tuning. Although the non-zero $z$-component of the DC electric field could cause spectral shift for ions at the edge of the waveguide with smaller Purcell enhancement, it is theoretically estimated to be more than an order of magnitude smaller than the electric-optic tuning (see Supplementary Note 5 for details). These results advance the dynamic control of REI-cavity coupling[15,16] down to single ion level. Combining with the ability to control ion frequency by Zeeman effect, full control over REI-cavity coupling system can be envisioned. The tunablity is also particularly useful when multiple REIs are to be addressed in a single cavity for frequency multiplexing, or integrating REIs coupled with different cavities on a single chip.

Although the decoherence and spectral diffusion property seen in ErLN is not as good as certain other hosts such as YSO[9], it can be greatly improved by applying external magnetic field[43]. Our previous work measured coherence time as long as 180 μs on smart-cut ErLN, under 0.5 T magnetic field and milli-kelvin temperature. This is already much longer than the cavity-enhanced lifetime in our resonator. Using a lower doping concentration will also help to improve coherence by suppressing Er-Er interaction[43]. Along with the capability to integrate on-chip microwave resonators, efficient manipulation of single rare earth spin is highly feasible based on our device. Another interesting perspective lies in the strong piezoelectricity of LN, as the effect of strain on REIs has been previously demonstrated[44]. The coupling between single REIs and mechanical mode[45] can be studied. With all these features, our device provides solid foundation for versatile and efficient REI-based spin-photon systems.

## Data availability

The data that support the findings of this study are included in the published article. Source data are provided with this paper.

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

## Acknowledgements

This work is supported by the US Department of Energy Co-design Center for Quantum Advantage (C2QA) under contract No. DE-SC0012704. We acknowledge initial funding from the Department of Energy, Office of Basic Energy Sciences, Division of Materials Sciences and Engineering under Grant DE-SC0019406. H.X.T. acknowledges partial support from the National Science Foundation (NSF) through ERC Center for Quantum Networks (CQN) grant EEC-1941583. The authors acknowledge helpful discussion with Dr. Charles W. Thiel and Dr. Baptiste Royer. The authors would like to thank Dr. Yong Sun, Kelly Woods, and Dr. Michael Rooks for their assistance provided in the device fabrication. The fabrication of the devices was done at the Yale School of Engineering & Applied Science (SEAS) Cleanroom and the Yale Institute for Nanoscience and Quantum Engineering (YINQE).

## Author contributions

H.X.T. and L.Y. conceived the experiment. L.Y. designed and fabricated the device. L.Y., S.W., M.S., and J.X. contributed to the preparation of the device. L.Y. and S.W. performed the experiment. L.Y. wrote the manuscript with contributions from all authors. H.X.T. supervised the work.

## Competing interests

The authors declare no competing interests.
