## [Peer Review File · Nature Communications]

Controlling single rare earth ion emission in an electro-optical nanocavityREVIEWER COMMENTS

Reviewer #1 (Remarks to the Author):

Probing single rare earth ions and controlling cavity-ion coupling is highly desirable in quantum networks. However, addressing the 4f-4f transition of single ions is challenging due to the long optical lifetime and resultant faint photoluminescence. In this work, the authors fabricate photonic crystal nanobeam cavities with high Purcell enhancement (177) and electro-optical tunability on smart-cut Er doped LNOI and demonstrate detection and control of single Erbium ion emission. Single Er ion emission is resolved and verified by g_2 measurement. Storage and retrieval of single ion excitation are realized by electro-optic tuning of the resonance frequency of the cavity while preserving the emission characteristic.

This is the first time that single Er emitters are studied in lithium niobate host. This work takes advantage of the high quality factor, small mode volume, and outstanding electro-optic property of integrated resonators on smart-cut thin film lithium niobate, paving way to scalable and controllable single photon sources and efficient spin-photon interfaces. I consider this work of high novelty and broad interest. The manuscript is easy to read and follow, the presented results are of high quality and discussed rigorously. Therefore, I would recommend this manuscript for publication if the authors could resolve the comments below:

1. A more careful and complete review of previous work on 1). Optical probing of single rare-earth ions emission and 2). electro-optical nanocavities in LNOI in the introduction section would be beneficial.
2. An energy levels diagram of Er ion in LNOI should be included, e.g., in Fig.3 (a).
3. In Fig.4, the authors investigate the storage and retrieval of single ion excitation at 10V and 80V. I am curious if the authors have tried other different voltages. It would be interesting to see how the lifetime could be extended as a function of the applied voltage. The authors observe a slightly lengthened lifetime at 80V (2.82ms) compared to waveguide value (2.5ms) and explain it as the modification of spontaneous emission rate in photonic crystal band gap. How large could the lifetime be extended if the applied voltage increases further?
4. Several necessary references are missing: In discussion section, the first sentence, for decoherence and spectral diffusion property of Er:YSO and for applying external magnetic field. In the fourth sentence, for using lower doping concentration to suppress Er-Er interaction.

Reviewer #2 (Remarks to the Author):

The manuscript demonstrates the active electro-optical control of single Er ion dopants embedded in a photonic crystal cavity patterned on thin-film lithium niobate. The results showcase an impressive Purcell enhancement allowing the authors to collect emission from single ions. The dynamic control of the emission rate by actively tuning the resonance position is particularly new and novel. The topic of controlling and collecting single photons from single rare-earth ions in nanophotonic structures is certainly of interest to the broad quantum optics and quantum information community. The manuscript is comprehensive and well-written, and I recommend it for publication with a few clarifications described below.

1. The authors demonstrate the use of an external electric field to modulate the optical properties of LN, which effectively allows active control of the resonance position. However, a discussion on how this electric field affects the emission of the Er ions is missing. How does the spectral shift of the ion emission compare with the cavity resonance shift while the voltage is applied? Does the applied voltage affect the spectral diffusion of the Er ions?
2. Fig. 4 (e) shows one nearly matching emission peak after the 100 us delay as described in the manuscript. However, the authors should explain why the rest of the spectrum looks so different.

Other minor recommended revisions:

1. Fig 4(a) and (c) use the term “Gate” to describe when the fluorescence signal is collected. The term can be confusing to the reader and the authors need to describe it more clearly in the main text. Is this purely in software or post processing or is it a hardware element? What was the duration of the gate in Fig 4(c)?
2. I would recommend making the x-label in Fig 2(c) “time” and not “delay” because it is a measured time, rather than an applied delay like the x axis in 4(d). In its current form, the similarity of 2(c) and 4(d) can be confusing.

Reviewer #3 (Remarks to the Author):

The manuscript reports on the fabrication of Er-doped lithium niobate nanophotonic devices. Purcell enhancement of Er ions coupled to the cavity is demonstrated. The authors also performed the electro-optic control of the Er ion emission in the nanophotonic cavity. The results show opportunities for controlling single emitters and spin-photon interfaces. However, the progress of physics and technology is too incremental to justify publication in Nature Communications.

Main points

1. The manuscript is similar to Nature communications¹², 1–7 (2021) (it is not probably cited in the Reference) and Optica 9, 445–450 (2022). The authors should discuss the difference and novelty in comparison with these works.
2. The authors emphasize the single ion detection in the manuscript. The second-order correlation function measurement shows a $g(2)=0.38$. Typically, $g(2)<0.5$ corresponds to a single photon emitter. It is under the assumption that all the emitters show similar fluorescence intensity. However, in the cavity, the enhancement of the emitters is different. The emitters show huge fluorescence intensity distribution. The $g(2)$ measurements of 0.38 cannot determine the single photon emitters in the cavity. A similar example can be found in Sci. Adv. 8 eabn9573 (2022).

Minor points:

1. P2 the authors mentioned that breakdown voltage of LNOI is larger than 500V. The breakdown voltage is a strong dependence on the distance between the electrodes. What is the gap size?
2. The authors should explain the reason for the nonlinear dependence of the EO tuning at the low voltage range in Fig. 1 (b).
3. Page 3, $T_{cav}=14$ us. What is the fitting function?
4. Fig. 4 (a) and (c) are difficult to understand. What is the gate used for?

In the end, the paper appears as some technical achievements, but the novelty in terms of physics and technique content is not obvious. If so, Nature Communications doesn't seem to be the appropriate journal to publish in. More technical journals should be considered.

We thank all reviewers for their thoughtful comments. In the following, we provide point-to-point responses to these comments. We have also revised the manuscript accordingly.

Response to Reviewer #1

Probing single rare earth ions and controlling cavity-ion coupling is highly desirable in quantum networks. However, addressing the 4f-4f transition of single ions is challenging due to the long optical lifetime and resultant faint photoluminescence. In this work, the authors fabricate photonic crystal nanobeam cavities with high Purcell enhancement (177) and electro-optical tunability on smart-cut Er doped LNOI and demonstrate detection and control of single Erbium ion emission. Single Er ion emission is resolved and verified by g_2 measurement. Storage and retrieval of single ion excitation are realized by electro-optic tuning of the resonance frequency of the cavity while preserving the emission characteristic.

This is the first time that single Er emitters are studied in lithium niobate host. This work takes advantage of the high quality factor, small mode volume, and outstanding electro-optic property of integrated resonators on smart-cut thin film lithium niobate, paving way to scalable and controllable single photon sources and efficient spin-photon interfaces. I consider this work of high novelty and broad interest. The manuscript is easy to read and follow, the presented results are of high quality and discussed rigorously.

We thank the reviewer for the valuable comments and his/her recommendation for the publication of this work. Below please find our point-to-point responses.

1. A more careful and complete review of previous work on 1). Optical probing of single rare-earth ions emission and 2). electro-optical nanocavities in LNOI in the introduction section would be beneficial.

We expand the discussion on optical probing of single rare earth ions as well as electro-optical nanocavities on LNOI in the introduction section of the manuscript. For single ion probing, we add one more recent paper on the use of Fabry-Perot cavity for detecting single Er ions (Ref. [1]).

2. An energy levels diagram of Er ion in LNOI should be included, e.g., in Fig.3 (a).

Thanks for the suggestion. We add an energy level diagram in Fig. 2c indicating that we pump at the Y_1-Z_1 transition (center around 1532nm) and collect resonant fluorescence. The corresponding figure caption is also revised.

3. In Fig.4, the authors investigate the storage and retrieval of single ion excitation at 10V and 80V. I am curious if the authors have tried other different voltages. It would be interesting to see how the lifetime could be extended as a function of the applied voltage. The authors observe a slightly lengthened lifetime at 80V (2.82ms) compared to waveguide value (2.5ms) and explain it as the

modification of spontaneous emission rate in photonic crystal band gap. How large could the lifetime be extended if the applied voltage increases further?

We thank the reviewer for putting forward the question. We did apply different voltages on another device whose resonance is at the center of Er inhomogeneous spectrum (we did not show these results in the manuscript because they are from Er ion ensembles instead of single ions). The results are shown in the figure below. The fluorescence intensity with respect to the delay is measured under different detuning voltage, shown by the triangular points. The curves are exponential fitting of the lifetime. It can be seen that the storage lifetime extends as the applied voltage increases.

Similarly, we also found that the lifetime saturates at ~ 2.8 ms for larger detuning voltage, as shown in the figure below with a voltage of 160V. We add a discussion on the saturation of lifetime at large detuning voltages in the manuscript. We consider the modified density of states due to photonic crystal band gap as a possible cause for slightly lengthened lifetime. It is also possible that the property of ions differs slightly at the tail of inhomogeneous distribution. The exact cause of this phenomenon and the specific role of band gap structure are subject to future studies.

4. Several necessary references are missing: In discussion section, the first sentence, for decoherence and spectral diffusion property of Er:YSO and for applying external magnetic field. In the fourth sentence, for using lower doping concentration to suppress Er-Er interaction.

We thank the reviewer for the suggestions. The discussion section now includes citation to Ref. [2] which presents narrow homogeneous linewidth of 73 Hz in Er:YSO, and Ref. [3] which discusses the coherence property of Er:LN with different concentration and external magnetic field.

Reference

[1] Ulanowski, Alexander, Benjamin Merkel, and Andreas Reiserer. "Spectral multiplexing of telecom emitters with stable transition frequency." *Science Advances* 8.43 (2022): eabo4538.

[2] Böttger, Thomas, et al. "Material optimization of Er³⁺: Y₂SiO₅ at 1.5 μm for optical processing, memory, and laser frequency stabilization applications." *Advanced Optical Data Storage*. Vol. 4988. SPIE, 2003.

[3] Thiel, C. W., et al. "Optical decoherence and persistent spectral hole burning in Er³⁺: LiNbO₃." *Journal of Luminescence* 130.9 (2010): 1603-1609.

Response to Reviewer #2

The manuscript demonstrates the active electro-optical control of single Er ion dopants embedded in a photonic crystal cavity patterned on thin-film lithium niobate. The results showcase an impressive Purcell enhancement allowing the authors to collect emission from single ions. The dynamic control of the emission rate by actively tuning the resonance position is particularly new and novel. The topic of controlling and collecting single photons from single rare-earth ions in nanophotonic structures is certainly of interest to the broad quantum optics and quantum information community. The manuscript is comprehensive and well-written, and I recommend it for publication with a few clarifications described below.

We thank the reviewer for the in-depth evaluation of our manuscript and positive comments. Below please find our point-to-point responses.

1. The authors demonstrate the use of an external electric field to modulate the optical properties of LN, which effectively allows active control of the resonance position. However, a discussion on how this electric field affects the emission of the Er ions is missing. How does the spectral shift of the ion emission compare with the cavity resonance shift while the voltage is applied? Does the applied voltage affect the spectral diffusion of the Er ions?

We thank the reviewer for the insightful comments. Theoretically, the DC stark shift of Er:LN vanishes when subjected to y-direction electric field. However, due to the non-uniformity of electric field, some ions might experience z-direction field which causes small spectral shift. This can be theoretically estimated with previous measured DC stark shift of $25\text{kHz}/\text{V}\cdot\text{cm}^{-1}$ [1] for z-direction electric field. The figure below shows electric field distribution in our lithium niobate waveguide for: a. optical mode; b. 1V DC tuning voltage; c. z-component field of 1V DC tuning voltage.

From the figure we can see that the z-component field at 1V voltage ranges from 0 at the center of the waveguide to $\sim 5 \times 10^4 \text{V/m}$ at the edge. This will result in a non-zero DC Stark shift of $\sim 10 \text{MHz/V}$ for ions that are on the edge who also experience less Purcell enhancement. The DC Stark shift should decrease toward zero for ions at the center with larger Purcell enhancement. Still, the value 10MHz/V is an order of magnitude smaller than the observed electro-optic tuning rate of 200MHz/V . We include this discussion in the Supplementary Information (Supplementary Note 5).

In our experiment, we were not able to directly assess the exact value of DC Stark shift of single ions for the following reasons: 1. The small frequency shift of fluorescence signal cannot be discriminated by the SNSPD; 2. The system drift (laser drift and spectral diffusion) prevents precision measurement of the DC Stark shift of single ions under different tuning voltage.

However, we did try an indirect measurement to estimate Er spectral shift in comparison with cavity resonance shift with ion ensemble in another device whose resonance is at the center of Er inhomogeneous spectrum. With fixed laser frequency and EO tuning voltage, we consider two different measurement schemes:

In scheme a, we first pump the ions on-resonance with the cavity, then detune the cavity by applying EO tuning voltage and collect the fluorescence decay to extract ion lifetime. Assume that the cavity detuning is Δ_c and the ion spectral shift caused by the voltage is Δ_i , then the ion-cavity detuning is $\Delta_c - \Delta_i$ during the fluorescence emission. For scheme b, the cavity is detuned by Δ_c before the pump pulse, while the probed ion frequency is constant at laser frequency, so the ion-cavity detuning is merely Δ_c . Since the Purcell enhancement (i.e., ion lifetime) depends on the ion-cavity detuning, we would expect to see different lifetime for two cases if Δ_i is significant enough. The measurement results are shown in the figure below.

The fact that these two curves almost overlap suggests that $\Delta_c - \Delta_i \approx \Delta_c$, or $\Delta_i \ll \Delta_c$, which matches well with our calculations.

As for the spectral diffusion under applied voltage, from Fig. 4e we can see that the linewidth of the peaks is similar for two curves. This suggests that the applied voltage does not cause significant change in spectral diffusion. But since the linewidth we measured is relatively broad, a detailed study should be done in the future with coherence measurement under magnetic field. We also add this discussion in the corresponding part of the manuscript.

2. Fig. 4 (e) shows one nearly matching emission peak after the 100 us delay as described in the manuscript. However, the authors should explain why the rest of the spectrum looks so different.

In Fig. 4e, the two spectrums are collected at separate time (~30 minutes apart). Thus, the system drift (~1MHz/min as estimated in the text) including laser drift and spectrum diffusion would come into play, causing random shift for different emission peaks. The emission peak used for storage measurement has a relatively small drift ~10MHz, while other peaks experience different amount of drift ranging from ~20 to 50MHz. We add a description of this behavior in the manuscript.

3. Fig 4(a) and (c) use the term “Gate” to describe when the fluorescence signal is collected. The term can be confusing to the reader and the authors need to describe it more clearly in the main text. Is this purely in software or post processing or is it a hardware element? What was the duration of the gate in Fig 4(c)?

Thanks for pointing out the confusion. “Gate” in Fig. 4a and 4c describes the time window the SNSPD is switched on for collecting fluorescence, thus is a hardware element. We clarify this point and add detailed definition of the labels “pump”, “gate”, and “EO voltage” in the caption of Fig. 4a. We also revise this portion of the text to define “gate” pulse as the control of SNSPD on/off. The duration of the gate in Fig. 4c is set to 20us and kept constant for different delay.

4. I would recommend making the x-label in Fig 2(c) “time” and not “delay” because it is a measured time, rather than an applied delay like the x axis in 4(d). In its current form, the similarity of 2(c) and 4(d) can be confusing.

We thank the reviewer for the suggestion. We modify the x-label in Fig. 2c from “delay” to “time”.

Reference

[1] Hastings-Simon, Sara R., et al. "Controlled Stark shifts in Er³⁺-doped crystalline and amorphous waveguides for quantum state storage." *Optics communications* 266.2 (2006): 716-719.

Response to Reviewer #3

The manuscript reports on the fabrication of Er-doped lithium niobate nanophotonic devices. Purcell enhancement of Er ions coupled to the cavity is demonstrated. The authors also performed the electro-optic control of the Er ion emission in the nanophotonic cavity. The results show opportunities for controlling single emitters and spin-photon interfaces. However, the progress of physics and technology is too incremental to justify publication in Nature Communications.

We appreciate the reviewer's thorough effort to evaluate our work and provide valuable feedback. We believe the publication in Nature Communications is well justified for the following reasons:

1) On the technology front, this work is **the first demonstration of Er:LN photonic crystal nanocavity** with ultrasmall mode volume ($\sim 0.15\lambda^3$) that offers strong Purcell enhancement. The advance to smart-cut LN platform with excellent optical properties offers opportunities for effective and active control of hosted emitters through electric field (as demonstrated here) and other forms of excitations (such as strain field), paving new ways for rare earth ion incorporated quantum technologies.

2) In physics point of view, this work is the **first to address single Er ions in LN host** and also the **first to demonstrate real-time control of rare earth emission on single ion level**. These results provide solid foundation for applications of such process in realizing controllable and efficient solid-state spin-photon systems.

We believe that our work is “**of high novelty and broad interest**” and “**of interest to the broad quantum optics and quantum information community**” as suggested by Reviewer #1&2 who recommend publication and is suitable for publication in Nature Communications.

1. The manuscript is similar to Nature communications¹², 1–7 (2021) (it is not probably cited in the Reference) and Optica 9, 445–450 (2022). The authors should discuss the difference and novelty in comparison with these works.

We agree that these two references ([1],[2]) are important related literatures and should be discussed with more emphasis in the manuscript. We revise the corresponding text to discuss these references more explicitly. We believe that our work has made substantial advances in terms of novelty and technical achievement compared to these works, on the following aspects:

1) Advancing to single ion detection and control. In our work, we advance the subject of study to single rare earth ions and verify the single ion property of emission with a second-order autocorrelation measurement. We further show that the cavity tunability can be used to shape and control single ion emission, promising new opportunities for controllable single photon sources and efficient spin-photon interfaces.

2) Device design and performance. Here, we fabricate photonic crystal nanobeam cavity with high quality factor and minimized mode volume. This allows more than 10-fold increase in Purcell enhancement, as a critical figure of merit, compared to the two references. Besides, the electro-optic tuning is also more definite and does not require active stabilization as compared to Ref. [1].

3) Material platform. Compared with Ref. [1], our device is based on thin-film lithium niobate platform that has excellent scalability and offers multifunctional capabilities (including but not limited to active electro-optic properties). The novel approach to integrate rare earth ions with smart-cut technique instead of ion implantation is also shown to better preserve bulk property as well as ensure full mode overlap, as compared to Ref. [2]. These advantages make our platform appealing for various quantum applications. The ability to incorporate Er emitters at telecom wavelength also promises new functionalities for lithium niobate integrated photonics.

2. The authors emphasize the single ion detection in the manuscript. The second-order correlation function measurement shows a $g(2)=0.38$. Typically, $g(2)<0.5$ corresponds to a single photon emitter. It is under the assumption that all the emitters show similar fluorescence intensity. However, in the cavity, the enhancement of the emitters is different. The emitters show huge fluorescence intensity distribution. The $g(2)$ measurements of 0.38 cannot determine the single photon emitters in the cavity. A similar example can be found in Sci. Adv. 8 eabn9573 (2022).

In the manuscript, we did consider the contribution from the background emitters that are weakly coupled to the cavity. We agree that a non-zero $g(2)$ cannot rule out the possible contribution from multiple emitters. However, this is not contradictory with our claim that the majority of collected photons are from a single emitter. Also, we believe that our case is best described by “a bright emitter with a background of weak emitters” as opposed to “two (or several) emitters with different brightness” for the following reasons:

1) The emission spectrum shows discrete peaks with continuous background, suggesting that the bright emitters are dilute enough to be resolved spectrally.

2) If the emission comes from two different emitters, their transition frequency will not be constantly same due to uncorrelated spectral diffusion (even if they have same center frequency). This will cause the emitted photons to split spectrally. However, in the $g(2)$ measurement, we observe a single emission peak with similar strength throughout the experiment, suggesting that it comes from a single emitter. This is also supported by the spectrum in Fig. 3c.

3) As discussed in Supplementary Note 4, the signal-to-noise analysis of $g(2)=0.38$ agrees well with the count rate we measure at the emission peak and the continuous background, indicating the single ion property of the emission peak.

Therefore, we believe that our results of single ion detection are valid. We add part of this discussion to Supplementary Note 4 to elaborate on this claim.

3. P2 the authors mentioned that breakdown voltage of LNOI is larger than 500V. The breakdown voltage is a strong dependence on the distance between the electrodes. What is the gap size?

For our device, the gap between electrodes is 5 μ m as indicated in Supplementary Note 1. We modify the manuscript to point out that the breakdown voltage of 500V is measured for this specific geometry.

4. The authors should explain the reason for the nonlinear dependence of the EO tuning at the low voltage range in Fig. 1 (b).

The imperfect linear behavior of EO tuning is likely due to the charge-screening effect, which manifests at cryogenic temperature in thin film lithium niobate. As discussed in Ref. [3], long relaxation time of the photorefractive effect of lithium niobate at cryogenic temperature could cause semi-permanent charge accumulation and redistribution from light illumination. These accumulated charges in lithium niobate could modify the device response to applied voltage and lead to nonlinear dependence on the tuning voltage.

5. Page 3, $T_{cav}=14$ us. What is the fitting function?

We thank the reviewer for raising the question. We use a single exponential fit for both waveguide and cavity cases. As discussed in Supplementary Note 3, we calculate the overall Purcell factor by averaging all ions in the cavity, weighted by their local field intensity. While the actual fluorescence decay should be a sum of all ions with different lifetime, we use a single exponential function to fit the averaged Purcell factor. A similar method was utilized in Ref. [4]. We modify the manuscript to clarify that we use a single exponential decay.

6. Fig. 4 (a) and (c) are difficult to understand. What is the gate used for?

We thank the reviewer for putting forward the concern. The gate pulse is used to control the on/off of the SNSPD, so that it can be switched off during pump pulse and switched on as fluorescence collection window. We revise the caption of Fig. 4a and the corresponding text to include definition of “pump”, “gate”, and “EO voltage” control sequences.

Reference

[1] Casabone, Bernardo, et al. "Dynamic control of Purcell enhanced emission of erbium ions in nanoparticles." *Nature communications* 12.1 (2021): 1-7.

[2] Xia, Kangwei, et al. "Tunable microcavities coupled to rare-earth quantum emitters." *Optica* 9.4 (2022): 445-450.

[3] Xu, Yuntao, et al. "Bidirectional interconversion of microwave and light with thin-film lithium niobate." *Nature communications* 12.1 (2021): 1-7.

[4] Miyazono, Evan, et al. "Coupling erbium dopants in yttrium orthosilicate to silicon photonic resonators and waveguides." *Optics express* 25.3 (2017): 2863-2871.

REVIEWERS' COMMENTS

Reviewer #1 (Remarks to the Author):

The authors have addressed all comments adequately. I recommend for publication.

Reviewer #2 (Remarks to the Author):

The authors answered all my questions and concerns satisfactorily, I recommend publication in Nature Comms.

Reviewer #3 (Remarks to the Author):

The authors have submitted a revised manuscript. The issues raised by myself as a referee have been carefully addressed and a detailed response has been supplied. I think that the manuscript is suitable for publication in Nat. Commun.